# Analysis and Experimental Study on Water Vapor Partial Pressure in the Membrane Distillation Process

**DOI:** 10.3390/membranes12080802

**Published:** 2022-08-19

**Authors:** Zanshe Wang, Zhaoying Jia, Ran Li, Qi Gao, Zhaolin Gu

**Affiliations:** School of Human Settlement and Civil Engineering, Xi’an Jiaotong University, Xi’an 710049, China

**Keywords:** membrane distillation, water vapor partial pressure, saturation pressure, Antoine’s equations, non-equilibrium state

## Abstract

In membrane distillation, the vapor pressure difference is the driving force of mass transfer. The vapor pressure is generally assumed by the saturation pressure and calculated by the Antoine equation. However, in the actual operation process, the feed solutions usually flow in a non-equilibrium state, which does not meet the theoretical and measurement conditions of the vapor-liquid equilibrium (VLE) state. This study tested the actual water vapor pressure of the pure water, lithium bromide (LiBr) solution, lithium chloride (LiCl) solution, and calcium chloride (CaCl_2_) solution under different flow conditions. The results showed that the actual water vapor pressure was lower than the saturation pressure overall, and the difference increased with temperature but decreased with the mass concentration. Therefore, in vacuum membrane distillation (VMD), air gap membrane distillation (AGMD), and sweeping gas membrane distillation (SGMD), the membrane flux calculated by water vapor saturation pressure was higher than the actual membrane flux, and the relative difference decreased and was less than 10% after 60 °C. In direct contact membrane distillation (DCMD), the water vapor pressure difference on both membrane sides was almost the same by using the saturation vapor pressure or the tested data since the pressure errors were partially offset in parallel flow or counter-flow modes.

## 1. Introduction

The membrane distillation process is widely used in separating aqueous solutions of non-volatile solutes. Four membrane distillation modes of vacuum membrane distillation (VMD), direct contact membrane distillation (DCMD), air gap membrane distillation (AGMD), and sweeping gas membrane distillation (SGMD) process are generally adopted according to the vapor treatment mode of the distillate side [1]. Due to the non-isothermal separation process, the vapor partial pressure difference between two sides of the membrane is the driving force of the mass transfer [2,3,4,5]. The membrane flux is generally described in Equation (1) [6,7,8]:(1)$$J=K_{m}(P_{f}-P_{p})$$
where *J* is the membrane flux, kg/(m^2^·s); *K*_m_ is the membrane transfer coefficient, s/m; *P*_f_ and *P*_p_ are the partial vapor pressures in the hot feed and distillate sides, Pa.

The membrane transfer coefficient *K*_m_ is the critical factor and a pretty complicated parameter related to the membrane material and its inherent properties [9] (e.g., membrane type, membrane construction, porosity, tortuosity, pore size, wall thickness, solution contact angle, the solution flow parameters). Much literature has been studied on membrane materials and their heat and mass transfer processes [10,11,12,13]. It is generally believed that the membrane transfer coefficient *K*_m_ was mainly decided by the inherent properties of membrane material and weak increases along with the temperature.

On the other hand, the vapor saturation pressure is one of the inherent properties of the aqueous solutions. Generally, based on the thermodynamic equilibrium states (thermal equilibrium, mechanical equilibrium, phase equilibrium, chemical equilibrium), the fundamental thermodynamic properties of the aqueous solution can be deduced and calculated by theory [14,15] and experimentally measured accurately [16,17]. As a result, the International Association for the Properties of Water and Steam (IAPWS) provided internationally accepted calculation formulas for water and steam properties and selected aqueous solutions [18,19]. Lange’s handbook of chemistry [20] is a well-documented and detailed chemical data handbook and reference book, which includes almost all known authoritative data on chemical elements, inorganic compounds, and organic compounds. The basic thermodynamic properties of the solutions by membrane distillation can be queried expediently. These thermodynamic properties guarantee convenience for analyzing and calculating membrane distillation. Therefore, the vapor saturation pressure of the aqueous solutions was usually identified as their vapor pressure. Accordingly, the mathematical expression of the vapor saturation pressure was derived from the Clausius–Clapeyron relation [10] and shown in the Antoine equation [21] of Equation (2). Most literature used the Antoine equation to calculate the vapor pressure of the aqueous solution [22,23,24,25,26,27,28,29].
(2)$$\log_{10}P=A-B/(t+C)\;$$
where *P* is the vapor saturation pressure, mmHg; *t* is the temperature, °C; A, B, and C are substance-specific coefficients. Different coefficients for different substances can be queried at the National Institute of Standards and Technology (NIST) chemistry webBook [30] and the Yaws handbook of vapor pressure [31].

It should be pointed out that the Antoine equation estimates the vapor saturation pressure was based on the aqueous solutions’ thermodynamic equilibrium state. However, in the actual operation condition of membrane distillation, the aqueous solutions are usually flowed and transported under the normal pressure condition. There was residual air in membrane pores and on the membrane surface. The physical condition of the ideal VLE state was far from being achieved, so the vapor and liquid states on the surface of the membrane were in a non-equilibrium state. Therefore, the saturation pressure assumption is not accurate either for the feed solution or for the driving force of the vapor pressure difference.

In membrane distillation, most feed solutions to be treated are mixed solutions containing various substances. Therefore, it is necessary to study the actual vapor pressure and the driving force pressure of the feed solutions under the non-equilibrium state.

Since most of the working fluids in the membrane distillation are aqueous solutions, in this study, pure water and hygroscopic salt solutions, such as the LiBr solution, the LiCl solution, and the CaCl_2_ solution were selected. The experimental tests were carried out in the flow state under normal pressure conditions in line with the actual state of the membrane distillation process as far as possible, and the results were contrasted with the vapor saturation pressure calculated by Antoine equations.

## 2. Materials and Methods

### 2.1. Vapor-Liquid Equilibrium State

The vapor-liquid equilibrium (VLE) state is the principal condition for the saturation pressure of aqueous solutions. Generally, the aqueous solution is placed in a vacuum container at a constant temperature. The volatile components begin to evaporate into a gaseous state. The gaseous molecules can also bounce back into the liquid state, and when vapor-liquid equilibrium is reached, the vapor saturation pressure of the volatile component is the vapor pressure of the solution.

In general, the theoretical derivation of the water vapor saturation pressure of the aqueous solutions was based on the VLE equations [32,33]. When an aqueous solution was in a VLE state, the Gibbs free energy of the liquid and its vapor was equal. Therefore, the temperature, the pressure, and the chemical potential of both the liquid and the gas phase were equivalent [14,15,32]. As a result, the Antoine equation was derived from the Clausius–Clapeyron relation of the solution under a VLE state.

### 2.2. Standard Measurement of Water Vapor Saturation Pressure

Strictly, the water vapor saturation pressure of the aqueous solutions was tested in the VLE apparatus [16,17,34], as shown in Figure 1. The test sample solution was placed into the glass ebulliometer, which was heated by the heating mantle, the vacuum pump controlled the operation pressure, and the water vapor was condensed in a spherical condenser and then returned to the mixing chamber for recirculation. The temperature and pressure were recorded after the VLE was reached [34]. The standard measurement of water vapor saturation pressure was tested under vacuum and gas-liquid equilibrium states.

### 2.3. The Saturation Pressure of the Aqueous Solution

Since the saturation pressure is an inherent property of aqueous solutions, the theoretical derivation and calculation of saturation pressure of the aqueous solutions were based on the VLE equations [33,34]. The authoritative and recognized analysis of saturation pressure of the aqueous solutions was more mature and accurate [35,36,37].

The pure water saturation pressure mainly focused on the Antoine equation (Equation (3)), the ASHRAE Fundamentals Handbook (Equation (4)), and the empirical formula (Equation (5)), as shown in Table 1.

The water vapor saturation pressure of the LiBr aqueous solution has been studied for many years, and many fitting calculation formulas were presented [48,49,50,51]. The widely used calculation formula was come from the ASHRAE Fundamentals Handbook [46] and shown in Equations (6) and (7):(6)$$\log P_{s}=C+D/T^{\prime }+E/T\prime^{2}$$
(7)$$T^{\prime }=(t-\sum_{0}^{3}B_{n}X^{n})/\sum_{0}^{3}A_{n}X^{n}+273.15$$
where *t* is the temperature of the aqueous solution, °C; *T*′ is the vapor temperature, K; *X* = %LiBr; *P*_s_ is the water vapor saturation pressure, kPa. *A*_i_, *B*_i_, *C*, *D*, and *E* are the coefficients shown in Table 2.

Similarly, the water vapor saturation pressure of the LiCl aqueous solution [52,53,54] and the CaCl_2_ aqueous solution [52,55,56] have been studied by many scholars, and the calculation formula was adopted the same expression, as shown in Equations (8)–(13), The difference lies in the regression coefficients shown in Table 3.
(8)$$P_{s}=\varepsilon (A+B\cdot T/T_{c})P_{H_{2}O}$$
(9)$$\varepsilon =1-{\left[1+{\left(X/\pi_{6}\right)}^{\pi_{7}}\right]}^{\pi_{8}}-\pi_{9}\cdot \exp[(-{\left(X-0.1\right)}^{2})/0.005]$$
(10)$$A=2-{\left[1+{\left(X/\pi_{0}\right)}^{\pi_{1}}\right)}^{\pi_{2}}$$
(11)$$B={\left[1+{\left(X/\pi_{3}\right)}^{\pi_{4}}\right)}^{\pi_{5}}-1$$
(12)$$P_{H_{2}O}=P_{c}\cdot \exp\left[\left(a\tau +b\tau^{1.5}+c\tau^{3}+d\tau^{3.5}+e\tau^{4}+f\tau^{7.5})/(1-\tau \right)\right]$$
(13)$$\tau =1-T/T_{c}$$
where *P*_s_ is the vapor saturation pressure, kPa; *T* is the temperature of the aqueous solution, K; *X* = %LiCl (or %CaCl_2_); *T*_c_ = 647.3 K; *P*_c_ = 22,090 kPa.

### 2.4. Non-Equilibrium State in Membrane Distillation

The calculation methods and the experimental test of the saturation pressure of the aqueous solutions based on VLE have been fully developed. Although there were slight differences in different calculation methods and test results, the testing data and the calculation methods provided a reasonable basis for analyzing and calculating the membrane heat and mass transfer process.

However, in the actual operating conditions, most of the feed solution flowed on the membrane side under normal pressure, and often in a turbulent state, air bubbles often appeared on the membrane surface, as shown in Figure 2a of our preliminary work [57]. Moreover, the membrane pores had residual air to influence the vapor diffusion, as shown in the membrane section in Figure 2b of our preliminary work [58].

Furthermore, since the average pore diameter of membrane material (membrane aperture) is usually comparable to the molecular mean free path, the diffusion process of vapor across the membrane was generally described as the Knudsen diffusion, the Molecular diffusion, or their combination [13]. Accordingly, the heat and mass transfer of the membrane can no longer be described by macro equilibrium thermodynamics, and it was in a non-equilibrium state.

Although the widely used assumption of water vapor saturation pressure *P*_f_ or *P*_p_ in an aqueous solution was reasonable to provide the basis and convenience for the analysis and calculation of membrane distillation, there would be some errors in the actual operating conditions owing to the non-equilibrium state. Therefore, analyzing and testing feed solutions’ real water vapor partial pressure in the actual flow state is meaningful and necessary.

## 3. Experiments

### 3.1. Method

Figure 3 shows the experimental apparatus for testing the water vapor partial pressure under the actual operation condition.

The test solution flowed through a horizontal aluminum tube with a diameter of 50 mm and a length of 400 mm, immersed in a constant temperature water bath (FDLDHX-2050, Nanjing, China, Accuracy: ±0.1 °C) to keep the stable temperature of the aqueous solution. A measuring hole with a diameter of 20 mm was arranged at the top of the aluminum tube, a 200 mesh 304 stainless steel screen was placed in the measuring spot to prevent splashing during the flow process, and a vertical aluminum tube with a diameter of 20 mm and a length of 120 mm was connected to the measuring hole. A temperature and humidity sensor (Rotronic HC2A-S, Bassersdorf, Switzerland: Pt100/A, −50–100 °C, accuracy: ±0.2 °C; Hygrometer HT-1, 0–100%Rh, accuracy: ±0.8%Rh) was installed in the vertical aluminum tube by the rubber stopper and placed in the air gap to test the temperature and the relative humidity of the wet air, which can figure out the actual water vapor pressure. There were a few air permeability pores in the rubber stopper to fit the operating conditions of membrane distillation. Two thermocouple sensors (Pt100, Shanghai, China, accuracy: ±0.2 °C) were installed separately in the aluminum tube and the water bath. The magnetic pump and the rotameter (LZB-25F, Changzhou, China, 250–2500 L/h; accuracy: ±1.5%) ensured the consistency of each flow measurement.

The actual water vapor pressure of the aqueous solution can be calculated by the tested temperature and the relative humidity. The feasibility of the test method lies in the following. First, wet air is a mixture, it is difficult to measure the water vapor pressure directly, but it can be calculated by measuring the air temperature and the relative humidity based on Dalton’s law of partial pressure [59]. Second, the membrane material was usually thinner, and there was no small enough probe and large enough space to arrange the sensors.

Consequently, the pure water, the LiBr solutions (40 wt%, 45 wt%, 50 wt%), the LiCl solutions (30 wt%, 35 wt%, 40 wt%), and the CaCl_2_ solutions (35 wt%, 40 wt%, 45 wt%) were put into the insulated tank (5 L), separately. The water bath temperature was set at a constant value from 30.0 °C to 85.0 °C. Each test was repeated three times, taking average values. The measuring hole and the temperature and humidity sensor were dried thoroughly before each test to ensure the independence test.

In order to verify the stability and the consistency of the experimental tests and the calculation formula, three measurements of pure water were carried out, and three calculation results of Equations (3)–(5) were presented, as shown in Figure 4. Three measurement results and three calculation results were in good agreement, separately. The red arrow represents the saturation pressures calculated based on Equations (3)–(5) of Table 1, the three formulas were in good agreement. The blue arrow represents the three measurement results, and the consistency of the three measurements was also perfect. Figure 4 indicates a difference between the vapor saturation pressure calculated by the Antoine equation and the actual measured data. The detailed analysis is presented in Section 3.3.

### 3.2. Uncertainty Analysis of the Experimental Measurements

There are only water vapor and air in the gas phase for pure water and the hygroscopic salt solution. Therefore, measuring the actual water vapor pressure indirectly was feasible by measuring the gas phase parameters. Firstly, the saturation pressure *P*_s_ at the temperature *T*_A_ state can be separately calculated by Equations (3), (6), and (8). Secondly, based on the relative humidity *φ*_A_ and *P*_s_, the actual water vapor pressure was calculated by the product of *P*_s_ and *φ*_A_, as shown in Equation (14).
(14)$$P=\varphi_{A}\cdot P_{s}=\varphi_{A}\cdot \exp[23.1964-3816.44/(T_{A}-46.13)]$$
where *φ*_A_ is the air relative humidity, % Rh; *T*_A_ is the air temperature, K; *P*_s_ is the water vapor saturation pressure, Pa; *P* is the actual water vapor pressure, Pa.

Since the parameters affecting test errors and uncertainties were the temperature *T*_A_, and the relative humidity *φ*_A_, the bias limit of the sensors is shown in Table 4. The error was uniformly distributed, and the inclusion factor *k* = $\sqrt{3}$. Then, the standard uncertainty of the test parameters can be calculated by u=Bias limit/k. Then, the standard uncertainty of the temperature was $u\left(T_{A}\right)=0.2/\sqrt{3}=0.115\;{}^{\circ}C$. The standard uncertainty of the relative humidity was $u\left(\varphi_{A}\right)=0.8/\sqrt{3}=0.462\;\%\mathrm{Rh}$. As a result, the combined standard uncertainty of the actual water vapor pressure *u*(*P*) was calculated by Equations (14) and (15). According to Equation (15), the standard deviation of the water vapor pressure was worked out and shown as error bars on the test results.
(15)$$u(P)={\left[P_{s}^{2}u^{2}(\varphi_{A})+\varphi_{A}^{2}u^{2}(P_{s})\right]}^{0.5}={\left[P_{s}^{2}u^{2}(\varphi_{A})+{\left(\varphi_{A}\cdot P_{s}\frac{3816.44}{{\left(T_{A}-46.13\right)}^{2}}\right)}^{2}u^{2}(T_{A})\right]}^{0.5}$$

### 3.3. Test Results and Discussion

For each test, the temperature setting of water bath *T*_0_ started from 30 °C and increased every 5 °C until 85 °C. The flow rates were kept at 20 L/min. Each test lasted one hour and was repeated three times, and the data acquisition system recorded the data every 3 s.

#### 3.3.1. Test Results

Figure 5a shows the test result of the pure water. The temperature of pure water and the wet air is shown on the left ordinate, and the actual water vapor pressure is calculated and shown on the right ordinate. Figure 5b shows the test results of the LiBr (45 wt%) solution, the LiCl (30 wt%) solution, and the CaCl_2_ (45 wt%) solution. The temperature of the aqueous solution and the wet air is shown on the left ordinate, and the relative humidity is shown on the right ordinate.

#### 3.3.2. Results Comparison

The actual measurement pressure and the water vapor saturation pressure calculated by Equations (3), (6) and (8) are shown in Figure 6. The legend in Figure 6 indicates the experimental and computed data separately, and the shaded part is the difference between the two sets of data.

Figure 6a shows that the saturation pressure of pure water was higher than the actual water vapor pressure, and the difference gradually increased with the temperature of the pure water.

Different from pure water, the LiBr, LiCl, and CaCl_2_ solutions are the liquid desiccant solution. They belong to the electrolyte salt solutions, with strong water absorption [60,61,62,63,64], so the higher the concentration of the solution, the lower the water vapor pressure. Figure 6b–d shows the actual water vapor pressure and the saturation pressure of the LiBr solutions, the LiCl solutions, and the CaCl_2_ solutions at different mass concentrations separately. Overall, the actual water vapor pressure was slightly lower than the saturation pressure *P*_s_, and the difference also increased with the increment of the solution temperature. Moreover, the difference (*P*_s_ − *P*_test_) decreased with the increment of the solution concentration.

#### 3.3.3. Discussion

According to the test results, the actual water vapor pressure under normal operating conditions was generally lower than the water vapor saturation pressure calculated by the VLE equations. The difference gradually increased with the increment of the feed solution temperature. The reasons are the following:

Firstly, as mentioned above, the calculation formulas of water vapor saturation pressure *P*_s_ by Equations (3), (6), and (8) were based on the Antoine equation derived from the VLE state. The water vapor saturation pressure was undoubtedly the maximum vapor pressure of the aqueous solution.

Secondly, the ideal measurement methods of VLE were rigorous. As mentioned in Section 2.2, the test solutions were first filled in a glass ebulliometer and at rest [17,34]. The vacuum pump pumped out the residual air. The aqueous solutions were heated to the boiling point to ensure the consistency of the temperature and pressure in the liquid and gas phases. The VLE process generally took a long time until gas-liquid equilibrium (Figure 7a). Therefore, the temperature of the liquid phase (*T*_L_) and gas phase were the same (*T*_g_). The water vapor saturation pressure (*P*_s_) was the maximum vapor pressure of the aqueous solution. However, in the actual operation conditions, the aqueous solution flows under the normal pressure, and the gas and liquid phase are in a non-equilibrium state. Besides, there is residual air in the membrane, and the volume of vapor molecules is relatively reduced. Owing to the mass transfer, the temperature of the gas phase (*T*_g_) is lower than that of the liquid phase (*T*_L_), as shown in Figure 7b. Therefore, the water vapor pressure (*P*_w_) under the actual operating conditions is only the surface pressure of the fluid and is lower than the saturation pressure (*P*_s_).

## 4. The Water Vapor Pressure in the Membrane Distillation Process

Because the real water vapor pressure was generally lower than the saturation pressure in the actual operating conditions, The water vapor transfer in the membrane distillation process will be slightly different from the traditional assumptions and calculation results. Therefore, the VMD, AGMD, and SGMD processes can be classified into one category because only one side of the membrane had an aqueous solution flowing. In contrast, the DCMD process can be classified into another category since the hot feed solution, and the cold solution flowed separately on two sides of the membrane, either with a parallel-flow mode or a counter-flow mode.

### 4.1. The Membrane Transfer Coefficient K_m_

The water vapor transfer process was usually described as the Knudsen diffusion, the molecular diffusion, or a combination [13,65]. Based on the hydrophobic polyvinylidene fluoride (PVDF) membrane material data of our preliminary work [66], the combination model was described in Equation (16). The calculation result of the membrane transfer coefficient *K*_m_ is shown in Figure 8.
(16)$$K_{m}={\left[\frac{3}{2}\frac{\tau \delta }{\varepsilon r}{\left(\frac{\pi RT}{8M}\right)}^{1/2}+\frac{\tau \delta }{\varepsilon }\frac{P_{1}}{PD}\frac{RT}{M}\right]}^{-1}$$
where *ε* = 0.85; *τ* = *ε*^−2.6^ = 1.526; *r* = 0.08 × 10^−6^ m; *δ* = 0.15 × 10^−3^ m [66]; *R* = 8.314 J/(mol·K); *M* = 0.018 kg/mol; *PD* = 1.895 × 10^−5^*T*^2.07^ Pa·m^2^/s [13]; *P*_1_ is the residual air pressure in pores, Pa.

The membrane transfer coefficient *K*_m_ weak increased along with the temperature, and the variation range is less than 5% within the temperature range of 60 °C. This variation range is smaller in the actual operation conditions because the temperature difference between the inlet and outlet of the membrane module is smaller than 60 K.

### 4.2. The VMD, AGMD, and SGMD Process

Figure 9a shows the schematic diagram of the heat and mass transfer in the VMD, AGMD, and SGMD processes. The liquid molecules on the aqueous solution side evaporated on the membrane pores, the vapor molecules (*m*_f_) passed through the membrane pores, and the latent heat was also transferred. On the other hand, the sensible heat (*Q*_f_) was conducted from the aqueous solution side to the distillate side. Since the distillate side generally keeps stable operating conditions, the water vapor pressure on the distillate side is decided by the vacuum degree, the temperature of the condensing plate, and the parameters of the sweep gas separately, which is constantly under stable conditions.

Accordingly, the water vapor pressure difference is qualitatively described in Figure 9b, where *P*_s_ is vapor saturation pressure, *P*_a_ is the actual vapor pressure, and *P*_p_ is the vapor pressure on the distillate side. The real water vapor pressure difference (*P*_a_ − *P*_p_) is lower than that of (*P*_s_ − *P*_p_), and the pressure error of Δ*P* = *P*_s_ − *P*_a_ gradually decreases with the direction of the aqueous solution flow because of the decrease in temperature and the increase in mass concentration, as shown in the hot-pink shadow in Figure 9b.

As a result, there will be some errors in the calculation process using saturation and actual pressure in the three membrane distillation processes. According to the test data and the calculated water vapor saturation pressure, Figure 10 shows the absolute error and relative error of 30 wt% LiCl solution, 45 wt% LiBr solution, and 45 wt% CaCl_2_ solutions in the operating temperature range of 303.15 K to 333.15 K. The absolute pressure error of Δ*P* increased with the solution temperature, as Figure 6b–d shows. However, since both the water vapor saturation pressure *P*_s_ and the tested water vapor pressure *P*_a_ increased exponentially with the temperature, while Δ*P* increased linearly (shown in Figure 10), the growth rate of Δ*P* is far lower than that of *P*_s_. Then, the relative error of Δ*P*/*P*_s_ declined with the solution temperature. Moreover, when the salt solution’s hygroscopicity is more robust, the increase of the absolute error is slight, and the decrease of the relative error is significant. The relative error is less than 10% at 60 °C (333.15 K) for the hygroscopic salt solution. For most of the aqueous solutions to be treated by membrane distillation, the operating temperature is higher than 60 °C (333.15 K), and the relative error (Δ*P*/*P*_s_) will rapidly decline below 10% after 60 °C, within an acceptable range at the membrane distillation operating temperature.

### 4.3. The DCMD Process

The schematic diagram of the water vapor pressure distribution in parallel-flow and counter-flow mode is qualitatively shown in Figure 11. In the DCMD process, the actual water vapor pressure difference (*P*_1a_ − *P*_2a_) is almost the same as that of the saturation water vapor pressure (*P*_1s_ − *P*_2s_) because the errors between the two sides can partially offset each other. Further instructions are shown in Figure 11b,d as the hot-pink shadow and the cyan shadow. That is, the hot-pink shadow represents the difference Δ*P*_1_ = *P*_1s_ − *P*_1a_ and the cyan shadow represents the difference Δ*P*_2_ = *P*_2s_ − *P*_2a_, Δ*P*_1_, and Δ*P*_2_ will partially offset each other in the DCMD process. In parallel-flow mode, the temperature difference on both sides of the membrane from the inlet end to the outlet end changes dramatically, and the absolute errors of Δ*P*_1_ > Δ*P*_2_ overall. Therefore, the fundamental errors on both sides of the membrane will partially offset, and the actual water vapor pressure difference will decrease slightly. As a result, the membrane flux will decrease slightly too. In counter-flow mode, the temperature difference on both sides of the membrane from the inlet end to the outlet end is relatively uniform. The absolute errors on both sides of the membrane will almost offset.

We quoted our previous simulation work on a hollow fiber membrane heat exchanger to describe the difference between the water vapor saturation and the actual water vapor pressure. A 55 wt% LiBr solution (inlet temperature of 100 °C, 1.63 L/min) and a 50 wt% LiBr solution (inlet temperature of 40 °C, 1.85 L/min) flowed in an 810 mm length of the hollow fiber membrane module [67], the membrane parameters were the same as those shown in Equation (16). Besides, the tested water vapor pressure of LiBr solution in this experiment was extended to 55 wt% mass concentration. Therefore, based on the calculation results and the testing data, the parameter distribution in the parallel-flow and the counter-flow mode is shown separately in Figure 12 and Figure 13. The temperature and the mass concentration of LiBr solution on both sides changed along the membrane length owing to the transmembrane heat and mass transfer of water vapor.

In the parallel-flow mode (Figure 12), due to the large temperature difference on both sides of the membrane in the inlet end, the water vapor pressure difference was significant, and the transmembrane transfer of water vapor mainly occurred at the beginning of the membrane component and then decreased rapidly to about 230 mm. After that, although the temperature on the hot side was still higher than that on the cooling side, the mass concentration was higher than that on the cooling side. As a result, the water vapor pressure difference between the two sides of the membrane was so tiny that the transmembrane transport of water vapor stopped. As the thermal conduction process continued, the water vapor pressure difference on both sides of the membrane was theoretically reversed (the vapor pressure curves intersect in Figure 12), but the mass transfer process of water vapor did not continue. On the other hand, on the hot side and the cooling side, the difference between the water vapor saturation pressure and the actual water vapor pressure was not much and almost offset each other in the calculation. According to the real water vapor pressure, although a short distance delayed the position of zero driving force on both sides of the membrane, it can be ignored for calculating the total membrane flux.

In the counter-flow mode (Figure 13), Since the temperature and water vapor pressure of the hot side were consistently higher than those of the cooling side, the transmembrane transfer of water vapor showed a uniform decrease in the direction of the membrane. Similarly, the difference between the water vapor saturation pressure and the actual water vapor pressure on the hot and cooling sides was offset.

### 4.4. Discussion

The water vapor pressure of the aqueous solution is the core parameter of membrane distillation, and its accuracy is of great significance to the performance analysis of membrane materials and the design of membrane components. The VLE-based Antoine equation and authoritative experimental data have provided a good and reliable data source for membrane distillation.

The aqueous solutions to be treated by membrane distillation are various and mainly belong to mixtures. In the actual flow process, they cannot meet the operating conditions of the theoretical derivation of vapor pressure and the precise experiment. As a result, there is a difference between the actual water vapor pressure and the water vapor saturation pressure, which will occur in errors in the calculation and analysis of the membrane distillation process.

Therefore, based on the non-equilibrium state, the experimental measurement of the actual water vapor pressure of pure water, the LiBr, LiCl, and CaCl_2_ solutions were carried out in this study. For the difference (Δ*P* = *P*_s_ − *P*_a_) between the water vapor saturation pressure calculated by the Antoine equation (*P*_s_) and the measured actual water vapor pressure (*P*_a_), the results showed that the difference of pure water was more significant. However, the error of liquid desiccant salt solution at an operating temperature above 60 °C will rapidly decline below 10%, which was acceptable.

In the VMD, AGMD, and SGMD processes, the driving force of water vapor pressure between two sides of the membrane would be lower. The effective way to improve the membrane flux is to increase the solution temperature by 1–2 °C at the same concentration based on the difference Δ*P*. In the DCMD process, the water vapor pressure difference was almost the same using the water vapor saturation pressure or the tested data calculation. The pressure errors partially offset each other in the parallel-flow and counter-flow modes.

Due to the transmembrane transfer of water vapor in membrane distillation, temperature polarization and concentration polarization typically exist on the membrane surface. That is, the temperature on the membrane surface is lower than that of the mainstream fluid, and the concentration on the membrane surface is higher than that of the mainstream fluid. The vapor pressure is the inherent property of the aqueous solution. Generally, the vapor pressure of the aqueous solution mainly depends on the temperature and concentration *P* = f(*T*, *X*), and the measurement data also show the same pattern. Therefore, on the membrane surface, the temperature decrease and concentration rise caused by temperature polarization and concentration polarization will follow the same calculation formula for saturated vapor pressure *P*_f_ = f(*T*_f_, *X*_f_) and actual vapor pressure *P*_m_ = f(*T*_m_, *X*_m_).

It is worth pointing out that the solution’s vapor-liquid equilibrium (VLE) state is a particular case of the vapor-liquid non-equilibrium state of the solution. The calculation of saturated vapor pressure based on VLE state in membrane distillation is a widely used reasonable assumption. This study agrees with using the water vapor saturation pressure in the membrane distillation process. However, the properties and parameters of the aqueous solutions are various, and a more accurate and in-depth study on membrane distillation should pay more attention to the non-phase equilibrium state of the aqueous solution.

## 5. Conclusions

This study compared the ideal VLE state of saturation pressure of aqueous solution and the non-equilibrium state of actual membrane distillation operation. The experimental tests of the pure water, the LiBr solutions, the LiCl solutions, and the CaCl_2_ solutions were carried out in the flow state in line with the actual condition in the membrane distillation process. The results showed that the actual water vapor pressure was lower than the saturation pressure overall, and the difference increased with the temperature but decreased with the mass concentration of the aqueous solution. In the VMD, AGMD, and SGMD processes, since only one side of the membrane has feed solution flowing, the membrane flux calculated by the saturation pressure was slightly higher than the actual membrane flux, but the relative error is less than 10% after 60 °C for the hygroscopic salt solution. In the DCMD process, the saturation vapor pressure calculation was still accurate because the difference between the saturation vapor pressure and the actual vapor pressure will almost offset each other on both sides of the membrane. It is still applicable to the concentration and temperature polarization in the membrane distillation since the actual water vapor pressure on both sides shows the same rule.

## Figures and Tables

**Figure 1 membranes-12-00802-f001:**
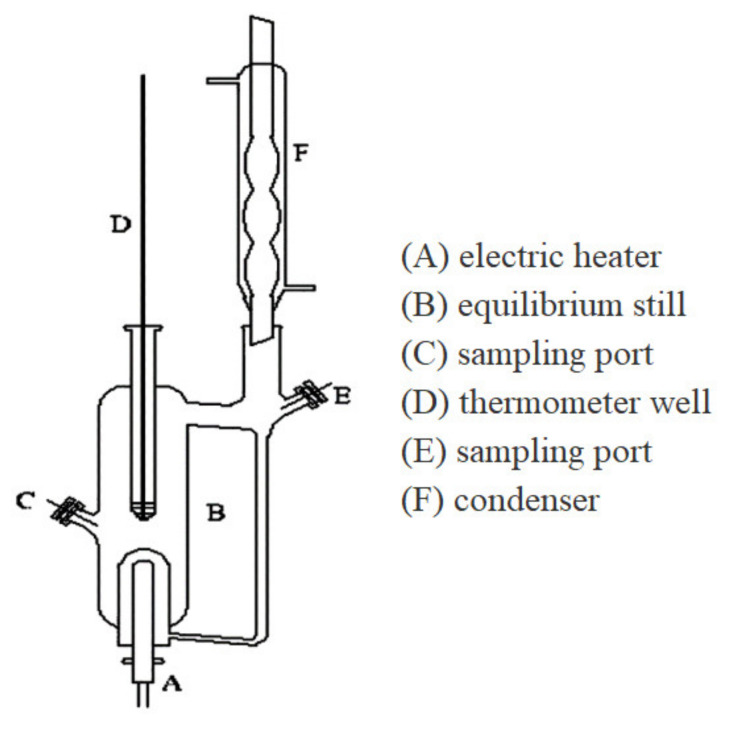
Scheme of VLE apparatus. Adopted with permission from Refs. [17,34], 2014, American Chemical Society.

**Figure 2 membranes-12-00802-f002:**
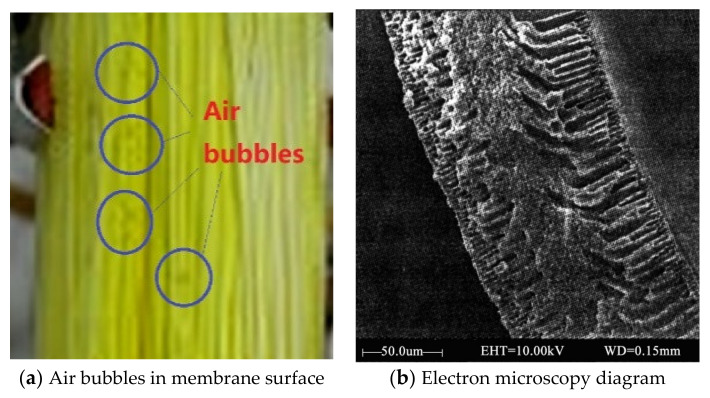
Membrane material and the actual operating condition.

**Figure 3 membranes-12-00802-f003:**
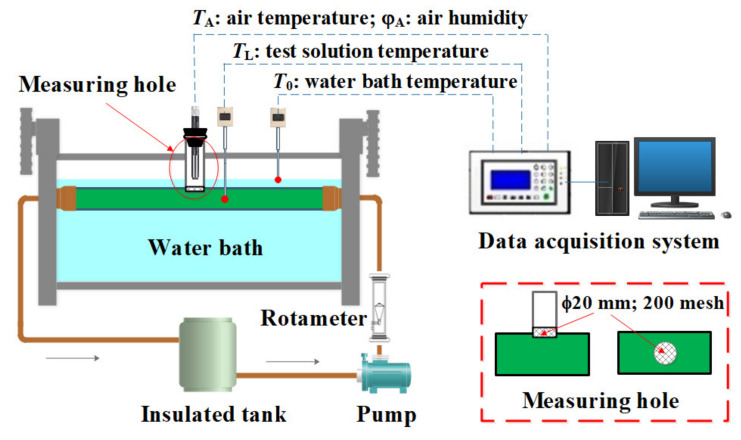
Flow chart of the test system.

**Figure 4 membranes-12-00802-f004:**
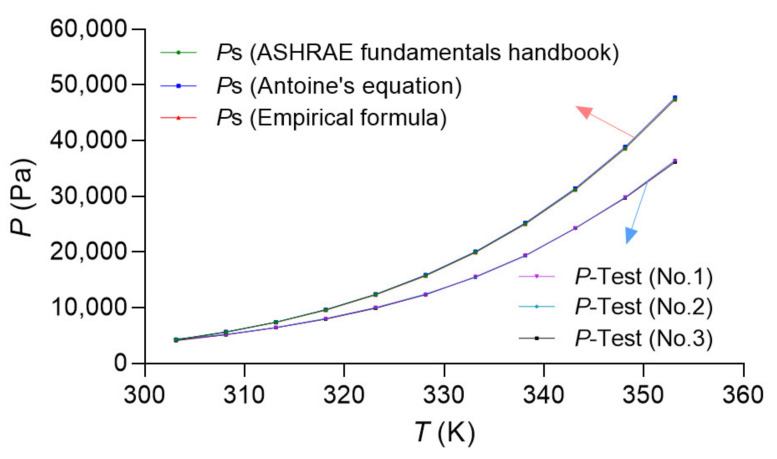
Three calculations and three tests result in pure water.

**Figure 5 membranes-12-00802-f005:**
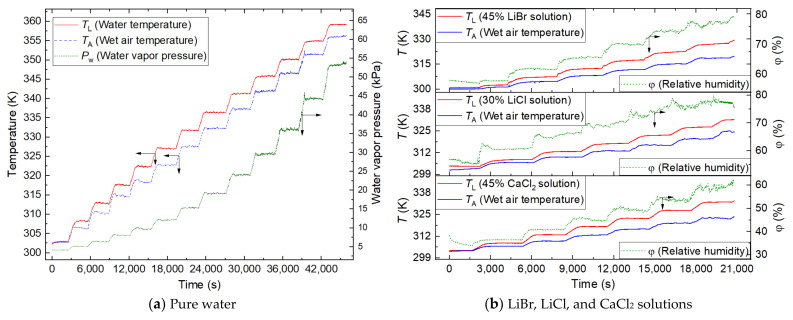
Test data of pure water and the hygroscopic salt solutions.

**Figure 6 membranes-12-00802-f006:**
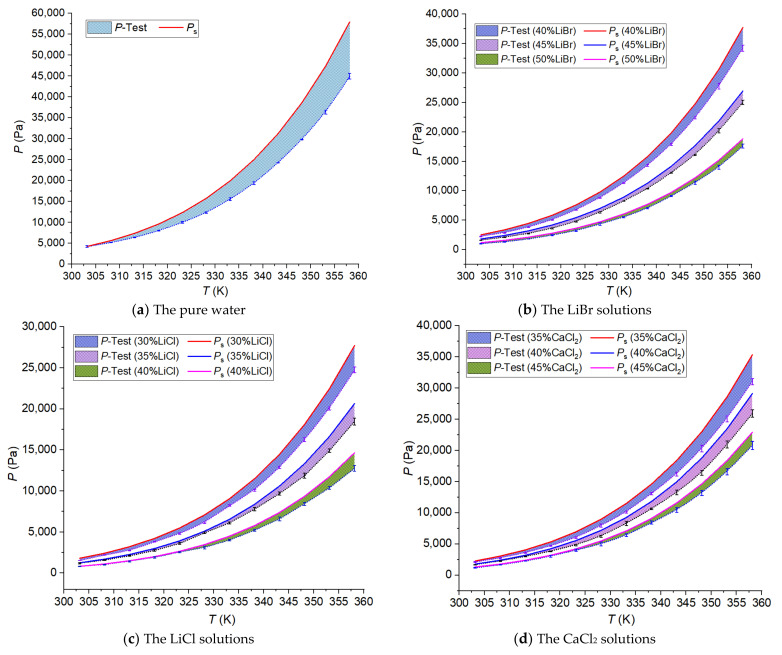
Comparison of water vapor pressure.

**Figure 7 membranes-12-00802-f007:**
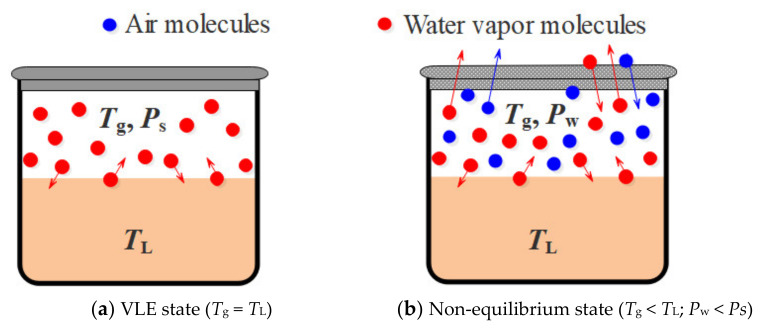
Schematic diagram of water vapor molecules in a different test state. *T*_L_: liquid phase temperature; *T*_g_: gas phase temperature; *P*_s_: water vapor saturation pressure; *P*_w_: water vapor actual pressure.

**Figure 8 membranes-12-00802-f008:**
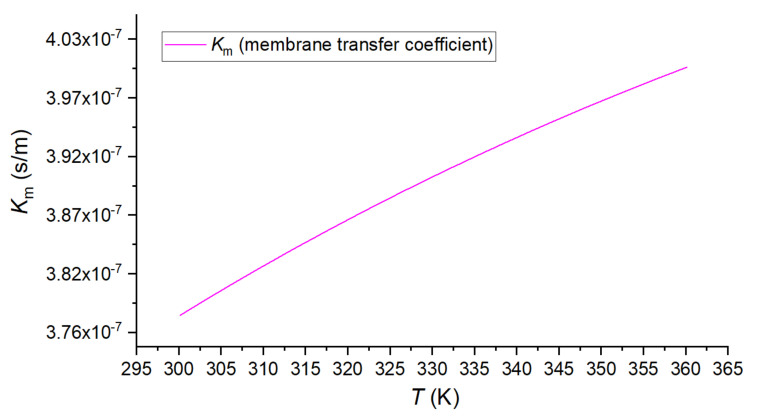
The membrane transfer coefficient *K*_m_ along with the temperature.

**Figure 9 membranes-12-00802-f009:**
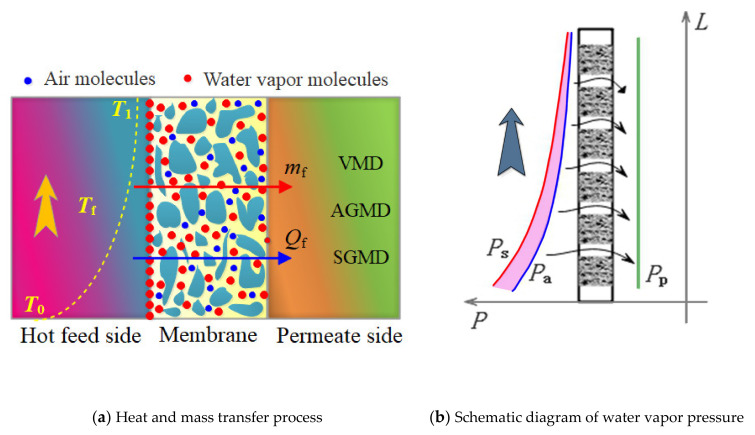
Schematic diagram of the VMD, AGMD, and SGMD process. In figure (**a**), *T*_0_: inlet temperature; *T*_1_: outet temperature; *T*_f_: solution temperature; *m*_f_: water vapor mass transfer; *Q*_f_: sensible heat transfer; VMD, AGMD, and SGMD represented the permeate side of three membrane distillation; The red arrow indicates the water vapor mass transfer process; The blue arrow indicates the sensible heat transfer process; In figure (**b**), the ordinate Y represents the length direction of the membrane (*L*); The abscissa X represents the water vapor pressure (*P*), and the black arrows indicate the direction of water vapor mass transfer; *P*_s_: vapor saturation pressure; *P*_a_: actual vapor pressure; *P*_p_: vapor pressure on the distillate side.

**Figure 10 membranes-12-00802-f010:**
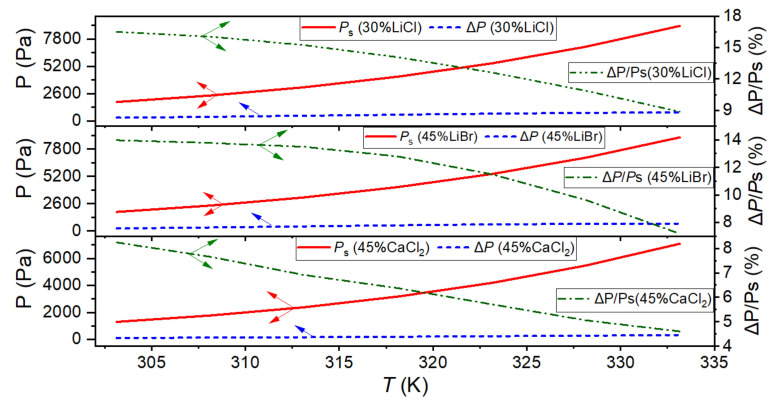
Absolute error and relative error of water vapor pressure.

**Figure 11 membranes-12-00802-f011:**
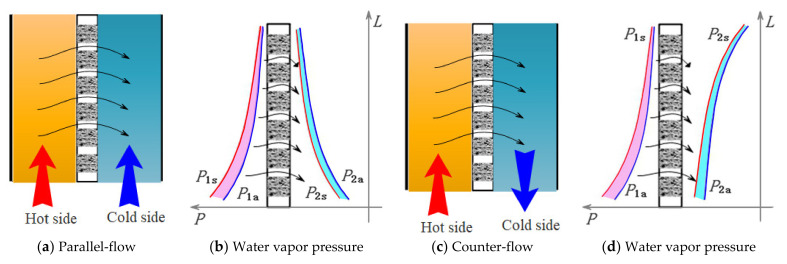
Schematic diagram of the DCMD process. In figure (**a**,**c**), red arrow indicates the direction of the hot side and blue arrow indicates the direction of the cold side, and the black arrows indicate the direction of water vapor mass transfer. In figure (**b**,**d**), the ordinate Y represents the length direction of the membrane (*L*); The abscissa X represents the water vapor pressure (*P*); *P*_1s_: vapor saturation pressure in hot side; *P*_1a_: actual vapor pressure in hot side; *P*_2s_: vapor saturation pressure in cold side; *P*_2a_: actual vapor pressure in cold side.

**Figure 12 membranes-12-00802-f012:**
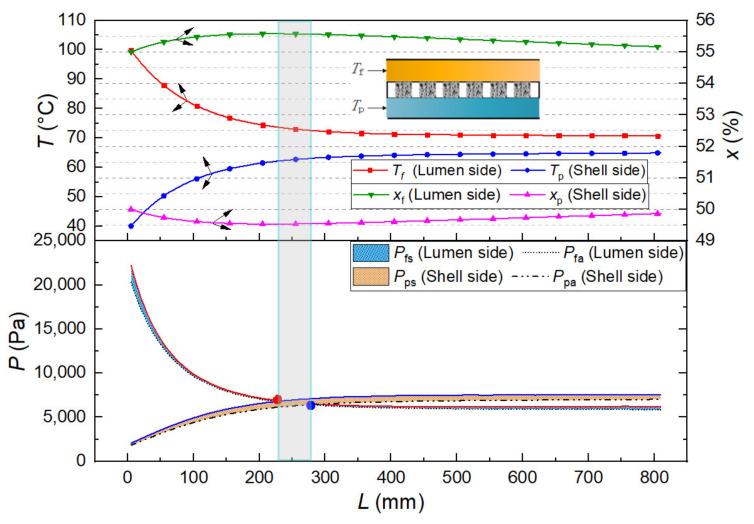
Parallel-flow mode of the DCMD process.

**Figure 13 membranes-12-00802-f013:**
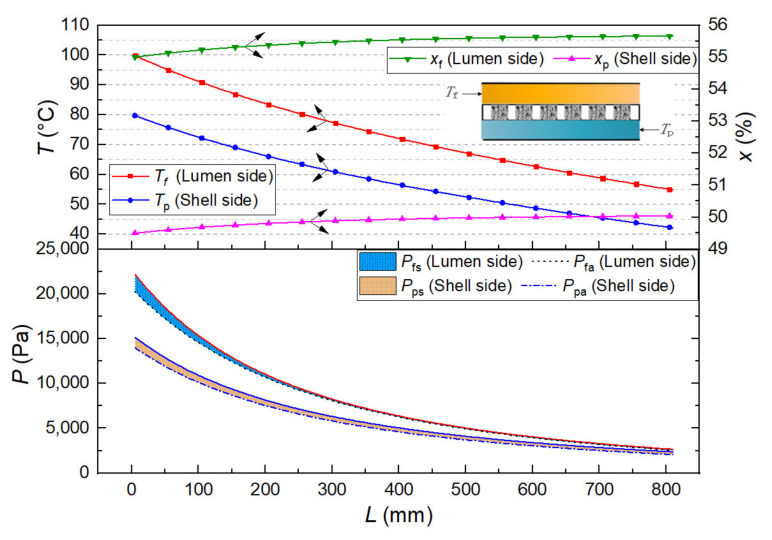
Counter-flow mode of the DCMD process.

**Table 1 membranes-12-00802-t001:** The water vapor saturation pressure of pure water.

Reference	Calculation Formula	
Antoine equation [38,39,40,41,42,43,44,45]	$P_{s}=\exp[23.1964-3816.44/(T-46.13)]$	(3)
ASHRAE Fundamentals Handbook [46]	$\ln P_{s}=A_{1}/T+A_{2}+A_{3}T+A_{4}T^{2}+A_{5}T^{3}+A_{6}\ln T$	(4)
	*A*_1_ = −5.8002206 × 10^3^; *A*_2_ = 1.3914993; *A*_3_ = −4.8640239 × 10^−2^;	
	*A*_4_ = 4.1764768 × 10^−5^; *A*_5_ = −1.4452093 × 10^−8^; *A*_6_ = 6.5459673	
The empirical formula [47]	$p_{s}=400/3\times \exp[18.5916-3991.11/(T-39.31)]$	(5)

**Table 2 membranes-12-00802-t002:** The coefficients in Equations (6) and (7).

i	0	1	2	3
*A* _i_	−2.00755	1.69760 × 10^−1^	−3.133362 × 10^−3^	1.97668 × 10^−5^
*B* _i_	1.24937 × 10^2^	−7.71649	1.52286 × 10^−1^	−7.9509 × 10^−4^
*C* = 7.05; *D* = −1.59649 × 10^3^; *E* = −1.040955 × 10^5^				

**Table 3 membranes-12-00802-t003:** The regression coefficients in Equations (8)–(13).

**π_0_**	**π_1_**	**π_2_**	**π_3_**	**π_4_**	**π_5_**
0.28/0.31 *	4.3/3.698 *	0.6/0.6 *	0.21/0.231 *	5.1/4.584 *	0.49/0.49 *
**π_6_**	**π_7_**	**π_8_**	**π_9_**	** *T* _c_ **	** *P* _c_ **
0.362/0.478 *	−4.75/−5.2 *	−0.4/−0.4 *	0.03/0.018 *	647.3 K	22,090 kPa
**a**	**b**	**c**	**d**	**e**	**f**
−7.85823	1.83991	−11.7811	22.6705	−15.9393	1.77516

* The regression coefficients of CaCl_2_.

**Table 4 membranes-12-00802-t004:** The bias limit and the standard uncertainty.

Parameters	*T*_A_ [°C]/Pt100/A	*φ*_A_ [%Rh]/Hygrometer HT-1
Bias limit	±0.2	±0.8
$u=\mathrm{Bias}\;\mathrm{limit}/\sqrt{3}$	0.115	0.462

## Data Availability

The data presented in this study are available on request from the corresponding author. The data are not publicly available since the data also forms part of an ongoing study.

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
