# Peer review of "Analysis and Experimental Study on Water Vapor Partial Pressure in the Membrane Distillation Process"

_membranes, 2022, doi:10.3390/membranes12080802_

Round 1

Reviewer 1 Report

There are still some questions not very clear. It is claimed that “although the absolute value of ΔP increases with the temperature, the relative error ΔP/Ps declines with the solution temperature increment. As shown in Fig. 10, the relative error is less than 10% at 60 ËšC (333.15 K) for the hygroscopic salt solution.”  ΔP is equal to Ps – Pa. When temperature increases, the absolute value of ΔP increases. This shows that Ps is increasing faster than Pa.

The relative error ΔP/Ps can be written as ΔP/Ps = (Ps – Pa)/Ps = 1-Pa/Ps. If Ps is increasing faster than Pa, 1-Pa/Ps should be larger. This means the relative error should be larger at higher temperature. Why does the author claim the relative error declines with temperature?

Another question is related to Figure 11. The feed side has higher temperature, while the permeate side has lower temperature. It is claimed that higher temperature results in a higher ΔP. Then, how could the ΔP1 and ΔP2 offset each other?

Author Response

We are particularly grateful to the editor and the reviewers for their careful reminders and valuable comments.

Please refer to the attachment for detailed reply. 

Best regards,

Wang zanshe

Reviewer 2 Report

Dear Authors

I liked your work very much, excellent.

A few small comments for what I consider to be necessary quantified descriptions

Line 12: ... water vapor pressure was a little less than… how much is a little less ?

Line 16: .. saturation pressure was slightly higher than… and how much is slightly?

Line 308: outlet of the membrane module is not high.  What does it mean: Is not high ?  Shouldn't it say here: ... smaller than, for example, 60 K

Line 399: .. .Therefore, some errors will occur in the membrane distillation process. Which errors occur here in the process?

Line 411 …on the the difference ΔP. In the DCMD process, the

Line 438: …the actual water vapor pressure was a little less than…   How much is a little less?

Author Response

(The authors gave the same response as above.)

Round 2

Reviewer 1 Report

The authors have addressed the questions satisfactorily. It is recommended for publication. 

This manuscript is a resubmission of an earlier submission. The following is a list of the peer review reports and author responses from that submission.

Round 1

Reviewer 1 Report

This work studied the ideal VLE state and the actual VLE state to compare the saturation pressure and actual vapor pressure. It was found that the ideal saturation pressure is higher than the actual vapor pressure. For salt solutions, the difference is smaller. Some more comments are:

1.     The MD application, salt water is usually used instead of pure water. How significant is the vapor pressure difference that affect the calculation?

2.     When the actual vapor pressure is smaller than the saturation, flux is smaller. How to solve this issue, and how to enhance the actual vapor pressure?

3.     For DCMD process, it is mentioned both sides of the pressure would offset each other. However, for DCMD, both sides are at different temperature. The author claims that the vapor pressure difference increases with the temperature. How could the two sides offset each other when they are at different temperature?

4.     It is claimed that “It is still applicable to the concentration polarization and temperature polarization in the membrane distillation”. How to justify this? How does the concentration polarization and temperature polarization affect the vapor pressure difference?

Reviewer 2 Report

I do not think this manuscript is qualified at this moment 

Reviewer 3 Report

In recent years, the descriptions of the MD process have been significantly simplified and as can be seen these generalizations create problems with the understanding of the MD process. As a result, the authors made incorrect assumptions and their work cannot be published, even after corrections, as this would perpetuate the literature confusion.

Equation (1) does not allow the analysis of the properties of membranes or flux in flow modules with housings - then we have three unknowns, the values of which are constantly changing along the module. Km can be determined for submerged modules, but only if we calculate PF and PD from the modeling. If these vapor pressures are read from temperature measurements, then Km includes a correction for temperature polarization and will vary with the temperature of the feed and its flow velocity. I doubt that immersing the enclosure in the bath (Fig. 3) would achieve the same inlet and outlet feed temperature - this required a careful balancing of the incoming and outgoing energy.

The assumption of non-equilibrium conditions adopted by the authors is also questionable - of course we have constant changes along the module. But we run the MD process in stationary conditions. On the surface of the membranes, we have a stationary liquid layer fed with a fixed heat flux. As a result, the vapor pressure on the segment dL is constant - it can be discussed what the length dL is. I believe that assuming 1-2 lengths of the free path will certainly not be error prone.

It should be taken into account that the use of salts further complicates the system due to concentration polarization - especially since these are salts that strongly lower the vapor pressure. The authors have implicated a process called osmotic membrane distillation (OMD) in all of this.

Fig. 11 – why the distribution of the vapor pressure is symmetrical - the conditions on the distillate side are always different from those on the feed side, hence symmetry is not possible.

Fig.12 – if the curves intersect, they still cannot diverge as the MD process stops. However, if we have non-isothermal conditions in the installation, then due to temperature changes along the length we activate conditions for the OMD process.